# Intriguing Properties of Adversarial Examples

## Abstract

It is becoming increasingly clear that many machine learning classifiers are vulnerable to adversarial examples. In attempting to explain the origin of adversarial examples, previous studies have typically focused on the fact that neural networks operate on high dimensional data, they overfit, or they are too linear. Here we show that distributions of logit differences have a universal functional form. This functional form is independent of architecture, dataset, and training protocol; nor does it change during training. This leads to adversarial error having a universal scaling, as a power-law, with respect to the size of the adversarial perturbation. We show that this universality holds for a broad range of datasets (MNIST, CIFAR10, ImageNet, and random data), models (including state-of-the-art deep networks, linear models, adversarially trained networks, and networks trained on randomly shuffled labels), and attacks (FGSM, step l.l., PGD). Motivated by these results, we study the effects of reducing prediction entropy on adversarial robustness. Finally, we study the effect of network architectures on adversarial sensitivity. To do this, we use neural architecture search with reinforcement learning to find adversarially robust architectures on CIFAR10. Our resulting architecture is more robust to white *and* black box attacks compared to previous attempts.

## 1 Introduction

An intriguing aspect of deep learning models in computer vision is that while they can classify images with high accuracy, they fail catastrophically when those same images are perturbed slightly in an adversarial fashion (Szegedy et al., 2013; Goodfellow et al., 2014). The prevalence of adversarial examples presents challenges to our understanding of how deep networks generalize and pose security risks in real world applications (Papernot et al., 2016a; Kurakin et al., 2016a). Several techniques have been proposed to defend against adversarial examples. Adversarial training (Goodfellow et al., 2014) augments the training data with adversarial examples. It has been shown that using stronger adversarial attacks in adversarial training can increase the robustness to stronger attacks, but at the cost of a decrease in clean accuracy (i.e. accuracy on samples that have not been adversarially perturbed) (Madry et al., 2017). Defensive distillation (Papernot et al., 2016b), feature squeezing (Xu et al., 2017), and Parseval training (Cisse et al., 2017) have also been shown to make models more robust against adversarial attacks.

The goal of this work is to study the common properties of adversarial examples. We calculate the adversarial error, defined as the difference between clean accuracy and adversarial accuracy at a given size of adversarial perturbation ($\epsilon$). Surprisingly, adversarial error has a similar dependence on small values of $\epsilon$ for all network models and datasets we studied, including linear, fully-connected, simple convolutional networks, Inception v3 (Szegedy et al., 2016), Inception-ResNet v2, Inception v4 (Szegedy et al., 2017), ResNet v1, ResNet v2 (He et al., 2016), NasNet-A (Zoph & Le, 2016; Zoph et al., 2017), adversarially trained Inception v3 (Kurakin et al., 2016b) and Inception-ResNet v2 (Tramèr et al., 2017), and networks trained on randomly shuffled labels of MNIST. Adversarial error due to the Fast Gradient Sign Method (FGSM), its L2-norm variant, and Projected Gradient Descent (PGD) attack grows as a power-law like $A\epsilon^B$ with $B$ between 0.9 and 1.3. By contrast, we find that adversarial error caused by one-step least likely class method (step l.l.) also scales as a power-law where $B$ is between 1.8 and 2.5 for small $\epsilon$. This observed universality points to a mysterious commonality between these models and datasets, despite the different number of channels, pixels, and classes present. Adversarial error caused by FGSM on the training set of

randomly shuffled labels of MNIST (LeCun & Cortes) also has the power-law form where $B = 1.2$, which implies that the universality is not a result of the specific content of these datasets nor the ability of the model to generalize.

To discover the mechanism behind this universality we show how, at small $\epsilon$, the success of an adversarial attack depends on the input-logit Jacobian of the model and on the logits of the network. We demonstrate that the susceptibility of a model to FGSM and PGD attacks is in large part dictated by the cumulative distribution of the difference between the most likely logit and the second most likely logit. We observe that this cumulative distribution has a universal form among all datasets and models studied, including randomly produced data. Together, we believe these results provide a compelling story regarding the susceptibility of machine learning models to adversarial examples at small $\epsilon$.

We show that training with single-step adversarial examples offers protection against large $\epsilon$ attacks (between 0.2 and 32), but does not help appreciably at defending against small $\epsilon$ attacks (below 0.2). At $\epsilon = 0.2$, all ImageNet models we studied incur 10 to 25% adversarial error, and surprisingly, vanilla NASNet-A (best clean accuracy in our study) has a lower adversarial error than adversarially trained Inception-ResNet v2 or Inception v3 (Kurakin et al., 2016b) (Fig. 1(a)). In light of these results, we explore a different avenue to adversarial robustness through architecture selection. We perform neural architecture search (NAS) using reinforcement learning (Zoph & Le, 2016; Zoph et al., 2017). These techniques allow us to find several architectures that are especially robust to adversarial perturbations. In addition, by analyzing the adversarial robustness of the tens-of-thousands of architectures constructed by NAS, we gain insights into the relationship between size of a model, its clean accuracy, and its adversarial robustness. In summary, the key contributions of our work are:

- We study the functional form of adversarial error and logit differences across several models and datasets, which turn out to be universal. We analytically derive the commonality in the power-law tails of the logit differences, and show how it leads to the commonality in the form of adversarial error.

- We observe that although the *qualitative* form of logit differences and adversarial error is universal, it can be *quantitatively* improved with entropy regularization and better network architectures.

- We study the dependence of adversarial robustness on the network architecture via NAS. We show that while adversarial accuracy is strongly correlated with clean accuracy, it is only weakly correlated with model size. Our work leads to architectures that are more robust to white-box and black-box attacks on CIFAR10 (Krizhevsky & Hinton, 2009) than previous studies.

## 2 Surprising Universality of Adversarial Error at Small $\epsilon$

FGSM computes adversarial examples as:

$$x^{\text{adv}} = x + \epsilon \, \text{sign} \left( \nabla_x L(x, y) \right), \tag{1}$$

where $x$ is the clean image, $y$ is the correct label for that image, $x^{\text{adv}}$ is the adversarial image, $\epsilon$ is the size of the adversarial perturbation, and $L(x, y)$ is the loss function. $\epsilon$ values are specified in range [0,255]. We only study white-box attacks in this section.

We begin with a preliminary examination of the architectural dependence of adversarial robustness. To that end, in Fig. 1 (a) we plot the test set adversarial error due to an FGSM attack as a function of $\epsilon$ for several models on ImageNet (Russakovsky et al., 2015). We note that for $\epsilon < 0.2$, the adversarial error follows a power law form with an exponent between 0.9 and 1.1 for all models studied. Even adversarially trained models Kurakin et al. (2016b), while adversarially much more robust for larger values of $\epsilon$, follow a similar form and reach as large as 20% adversarial error at smaller $\epsilon$.

In light of the surprising commonality in adversarial error at small-$\epsilon$, we investigate whether there is any way to get a different form for the adversarial error. To do this, we evaluate the adversarial error due to step l.l. attack, which is computed as:

$$x^{\text{adv}} = x - \epsilon \, \text{sign} \left( \nabla_x L(x, y_{l.l.}) \right), \tag{2}$$

where $y_{l.l.}$ is least likely class predicted by the network on clean image $x$ (Kurakin et al., 2016b). The adversarial error also follows a power law, however with a larger exponent. The exponents range from 1.8 to 2.2 for ImageNet models (Fig. 1(b)), and 1.8 to 2.5 for models trained on MNIST (Fig. 2(c)) and CIFAR10. Thus, we see that attack protocol can change the exponent of the observed power-law.

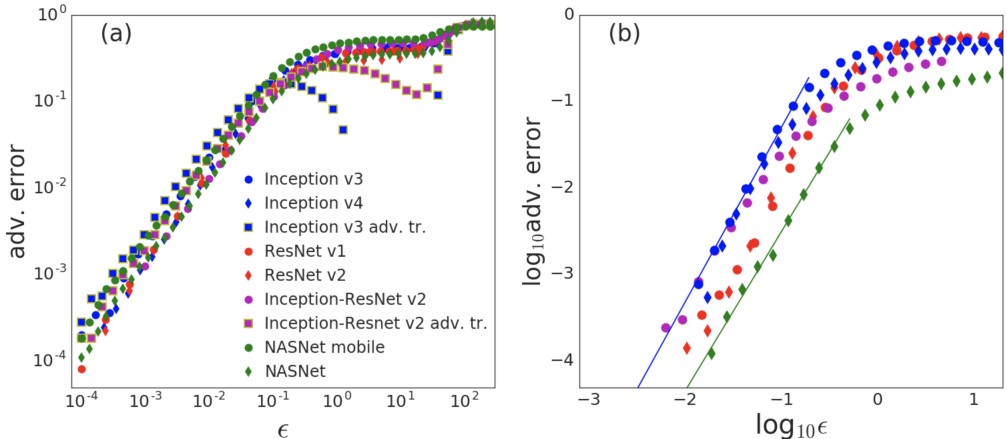

Figure 1: Test set adversarial error as a function of $\epsilon$ for models trained on ImageNet due to FGSM and step l.l. attack in (a) and (b), respectively. adv. tr. denotes models that are adversarially trained (Kurakin et al., 2016b). In (b), we also show two of the power law fits with straight lines.

To test the limits of the universality observed in Fig. 1, we perform a number of more extensive tests. First, we investigate the effect of architecture by stochastically sampling thousands of different neural networks and train them on MNIST. We then measure their adversarial error due to FGSM on the test set. The architectures we sample are either fully-connected networks with 1-4 hidden layers and 30-2000 hidden nodes in each layer, or simple convolutional networks with dropout rates between 0-0.5. The adversarial error of representative linear, fully-connected, and convolutional networks are shown in Fig. 2 (a). As above, these models all have the same form of adversarial error with a powerlaw dependence on $\epsilon$ with exponents between 0.9 and 1.2.

See Fig. 9 in the Appendix for a plot with all of the generated networks. We perform the same analysis on a 32-layer ResNet trained on CIFAR10 (He et al., 2016), which achieves a 92.6% clean accuracy on the test set. The result is shown in Appendix Fig. 10, where the adversarial error follows a power law with an exponent of 0.99 up to an $\epsilon$ of 1.

Next, we probe the relationship between generalization and adversarial robustness following a similar approach to Zhang et al. (2016). In particular, we train a fully connected network on MNIST with shuffled labels until it reaches perfect accuracy on the training set. The adversarial error on the training set is shown in Fig. 2(b). Once again we see that the adversarial error follows an almost identical power-law form at small $\epsilon$ with an exponent of 1.2.

Finally, we further investigate the dependence of adversarial robustness on attack protocol. We plot in Fig. 2 (c) the adversarial error on MNIST with an $L_\infty$-normalized FGSM attack, an $L_2$-normalized FGSM attack, and a 20-step projected gradient descent (PGD) attack (Madry et al., 2017). We see that despite the anomalous exponent observed for step-l.l. attacks, the other attack methods display the same universality with exponents of 1.1, 1.2, and 1.3 for L2-norm, FGSM, and PGD attacks, respectively. step-l.l. attack on MNIST has an exponent of 2.3.

## 3 A MEAN-FIELD THEORY OF ADVERSARIAL PERTURBATIONS

### 3.1 LINEAR RESPONSE

We now offer a theoretical explanation for the observed universal behavior of adversarial error. The breadth of these observations shows that adversarial error for small adversarial perturbations

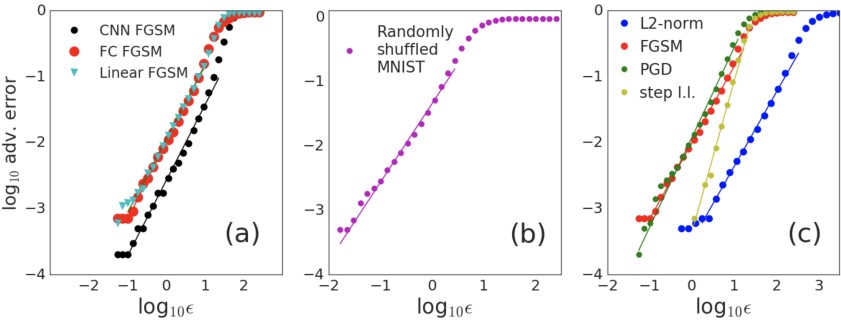

Figure 2: Points represent adversarial error as a function of $\epsilon$ for models trained on MNIST. Straight lines are power-law fits. (a) FGSM attack on fully-connected 3-layer network (FC), a linear model, and a convolutional network (b) FGSM attack on FC trained on randomly shuffled labels (evaluated on the training set). (c) Different attacks on FC.

does not depend on the specifics of the neural network, which implies that we can understand the small $\epsilon$ regime by making simplifying approximations. We begin by considering the linear response of a neural network to adversarial perturbations. Another approach to adversarial examples that considers the linear response of the network can be found in Nayebi & Ganguli (2017). The effect of margins on adversarial robustness has been brought up in **?**.

We will study an $L_2$-variant of the FGSM attack. Here, the adversarial perturbation is given by,

$$x^{\mathrm{adv}} = x + \epsilon \frac{\nabla_x L}{||\nabla_x L||_2} \tag{3}$$

instead of the more commonly used $\ell_\infty$ variant. As shown above, the form and exponent of adversarial error is qualitatively insensitive to this choice (see Fig. 2(c)). We will now attempt to compute the minimum $\epsilon$, that we call $\hat{\epsilon}(x)$, required before the class assigned to an input, $x$, changes. Assuming the network was able to perfectly classify clean images, the adversarial error rate will then be $P(\hat{\epsilon} < \epsilon)$. While perfect classification will not be achieved in practice, the insensitivity of the form of adversarial error to clean accuracy demonstrated above for many systems suggests that this approximation is sound.

Notationally, we will refer to the output of the network as $\hat{y}_i(x)$ and the corresponding logits as $h_i(x)$. The class prediction of the network will then be $\mathrm{argmax}(\hat{y}(x)) = \mathrm{argmax}(h(x))$. For simplicity we will choose an ordering of the logits such that $h_1(x) \geq h_2(x) \geq \cdots \geq h_N(x)$. We can then enumerate a set of logit-differences, $\Delta_{ij}(x) = h_i(x) - h_j(x)$. If an adversarial perturbation is to successfully cause the network to make an erroneous prediction, then it must be true that $h_1(x^{\mathrm{adv}}) < h_j(x^{\mathrm{adv}})$ for at least one $j$.

We calculate the response of the logits to adversarial perturbation. We consider the linearized response of the network and find that in the limit of small $\epsilon$ (see Appendix 6.2.1),

$$h(x^{\mathrm{adv}}) = h(x) + \epsilon \frac{J^T J \delta}{||J\delta||_2} + \mathcal{O}(\epsilon^2) \tag{4}$$

where $J_{ij} = \partial h_j / \partial x_i$ is the input-logit Jacobian of the network and $\delta_i = \partial L / \partial h_i$ is the error of the outputs of the network. For notational convenience we will define $\Gamma(x) = J^T J \delta / ||J\delta||_2$. In this linear model we therefore predict that the logit-differences will scale as follows,

$$\Delta_{ij}(x^{\mathrm{adv}}) \approx \Delta_{ij}(x) + \epsilon(\Gamma_i(x) - \Gamma_j(x)) + \mathcal{O}(\epsilon^2). \tag{5}$$

Recall that the adversarial perturbation will successfully cause the network to just barely misclassify an input precisely when $\Delta_{1j}(x^{\mathrm{adv}}) = 0$ for at least one $j$. We can predict per-class $\epsilon$-thresholds beyond which the network will misclassify a given point,

$$\hat{\epsilon}_j(x) = \frac{\Delta_{1j}(x)}{\Gamma_j(x) - \Gamma_1(x)}. \tag{6}$$

Together this allows us to compute a linear approximation to $\hat{\epsilon}$ given by $\hat{\epsilon}_{\text{linear}}(x) = \min_j (\epsilon_j(x))$.

We can confirm that the change in the logits for small changes in the inputs is well-described by this linear model. This is shown in fig. 3 (a) where we see the logits for a single example upon

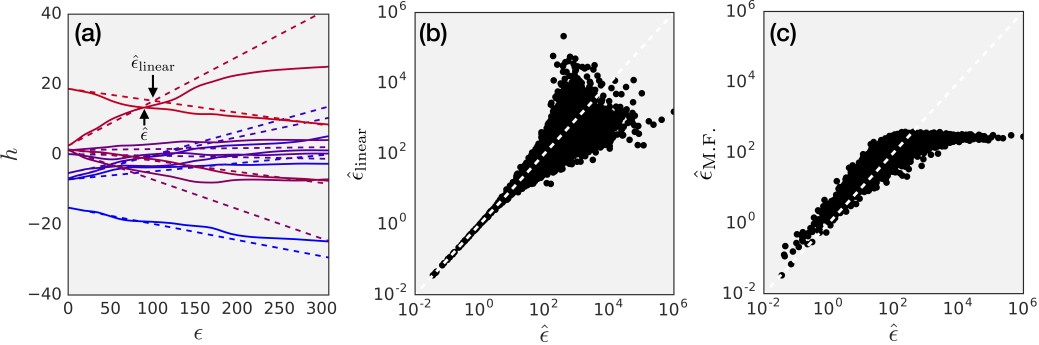

Figure 3: Linear approximation for the response of the logits to adversarial perturbation. (a) The dynamics of logits to an adversarial perturbation as a function of size $\epsilon$ for a single training example. Dashed lines show the linear approximation. Colors indicate the ranking of the logit from largest (red) to smallest (blue). (b) The smallest $\epsilon$ needed to fool the network ($\hat{\epsilon}$) for individual test examples compared with the prediction from the linear theory. (c) Mean field prediction of $\hat{\epsilon}$.

perturbation over a range of $\epsilon$. In particular, for small $\epsilon$ we see an excellent agreement between the linear approximation and the true logit dynamics. We also see that the $\epsilon$ where the first and second logit cross is well-approximated by the linear prediction. In Fig. 3 (b) we plot $\hat{\epsilon}_{\text{linear}}(x)$ against $\hat{\epsilon}(x)$ evaluated on every MNIST example in the test set. The white dashed line is the line $\hat{\epsilon}(x) = \hat{\epsilon}_{\text{linear}}(x)$. We see that when $\hat{\epsilon}(x)$ is small the $\hat{\epsilon}_{\text{linear}}(x)$ concentrate increasingly around the $\hat{\epsilon}(x)$. Together these results show that the linear response predictions are valid for small adversarial perturbations.

While the linear model outlined above gives excellent agreement in the $\epsilon \to 0$ limit, the $\Gamma_i(x)$ are themselves complicated objects (being functions of the Jacobian). This makes the analytic evaluation of Eq. (6) challenging. We therefore introduce a "mean-field" approximation to Eq. (6) by replacing $\Gamma_i(x)$ by its average over the dataset, $\langle \Gamma_i(x) \rangle$. Similar independence approximations have previously been successful in analyzing the expressivity and trainability of neural networks (Schoenholz et al., 2016; Poole et al., 2016). Finally, we observe that the vast majority of the time (for example, more than 95% of successful FGSM attacks for $\epsilon < 50$), it is $\Delta_{12}(x^{\text{adv}})$ that goes to zero before any of the other $\Delta_{1j}$. We therefore assume that this will be the dominant failure mode for neural networks and write down a mean-field estimate for $\hat{\epsilon}$,

$$\hat{\epsilon}_{\text{M.F.}} = \frac{\Delta_{12}(x)}{\langle \Gamma_2 \rangle - \langle \Gamma_1 \rangle}. \tag{7}$$

We show in Fig. 3 (c) that this approximation continues to be strongly correlated with $\hat{\epsilon}$. Together these results suggest that the adversarial error rate for perturbations of size $\epsilon$ will be

$$P(\hat{\epsilon} \leq \epsilon) \approx P\left[\Delta_{12} \leq \epsilon(\langle \Gamma_2 \rangle - \langle \Gamma_1 \rangle)\right] = P(\Delta_{12} \leq \tilde{\epsilon}) \tag{8}$$

where we have defined $\tilde{\epsilon} = \epsilon(\langle \Gamma_2 \rangle - \langle \Gamma_1 \rangle)$ to be a network-specific rescaling of $\epsilon$. We therefore expect the adversarial error rate at small $\epsilon$ to be dictated by the cumulative distribution of $\Delta_{12}$ for attacks that effectively target the second most likely class (e.g. FGSM, PGD ...etc.).

### 3.2 Universal Properties of the Logit Difference Distribution

With the results from the preceding section in hand, we now investigate the distribution of logit differences, $P(\Delta_{1j})$. Since we are particularly interested in the small $\epsilon$ regime, we seek to compute $P(\Delta_{1j})$ for small $\Delta_{1j}$. To make progress we will again make a mean field approximation and assume that each of the logits are i.i.d. with arbitrary distribution. With this approximation we find that for small $\Delta_{1j}$ (see Appendix 6.2.2),

$$P(\Delta_{1j}) = C \Delta_{1j}^{j-2} + \mathcal{O}(\Delta_{1j}^{j-1}) \tag{9}$$

where $C$ is a network specific constant. An interesting consequence of this result is that the difference that we are particularly interested in, $P(\Delta_{12})$, scales as $\mathcal{O}(1)$ as $\Delta_{12} \to 0$. This implies that, generically, we expect the most likely logit and second most likely logit to have a finite probability of being arbitrarily close together. We interpret this as an inherent uncertainty in the predictions of neural networks.

While it is not obvious that the assumption of a factorial logit distribution is valid here, we will see that Eq. (9) captures the universal features of the distribution at small values of the logit difference. Indeed, from the previous section we see that the form of Eq. (9) implies that the adversarial error rate should scale as follows

$$P(\hat{\epsilon} < \epsilon) \approx P(\Delta_{12} < \tilde{\epsilon}) \approx C\tilde{\epsilon} + \mathcal{O}(\tilde{\epsilon}^2) \tag{10}$$

as was broadly observed in the previous section.

To further test whether or not the mean field approximation is valid, we evaluate $\Delta_{1j}$ for a number of different neural network architectures and datasets. We find that on all datasets and models studied, the $\Delta_{1j}$ distributions have power-law tails. As predicted, $\Delta_{12}$ has a power-law tail with an exponent of about 0, and $\Delta_{1j}$ for $j > 2$ have power-law tails with positive exponents increasing with $j$. We note, however, that the powers are typically not integral for large $j$. It seems likely that this breakdown is the result of correlations between the logits. In Fig. 4, we compare distribution of $\Delta_{1j}$ with $j = 2, 3, 4$ for ImageNet and logits that are independently sampled from a uniform distribution for 5 million samples with 10 classes. In Fig. 6 we see similar results for MNIST. Together these results verify our predictions over a vast set of networks and datasets.

The empirical results above show that adversarial error has a power-law form for well-studied and random datasets, simple full connected networks as well as complicated state-of-the-art models. It follows that the prevalence and commonality of adversarial examples is not due to the depth of the model (for example, see the linear model in Fig. 2 (a)), or the high-dimensionality of the datasets. Rather, they are due to the fact that lots of examples have small $\Delta_{12}$ values. This makes it easy to find examples to fool the model at test-time.

It is interesting that the distribution of $\Delta_{1j}$, at small $\Delta_{1j}$, for trained models is essentially identical to that of i.i.d. random logits (especially for $j = 2$). This suggests that while our training procedures are good at modifying the largest logit in a way that leads to good clean accuracies, these procedures do not induce strong enough correlations between the logits to disrupt the essential scaling uncovered above. This problem is reminiscent of the problem distillation (Hinton et al., 2015) attempts to solve, by incorporating information about ratios of incorrect classes. This might be one of the reasons defensive distillation improves adversarial robustness. It would be interesting to study the distributions of logit differences during training of distillation networks.

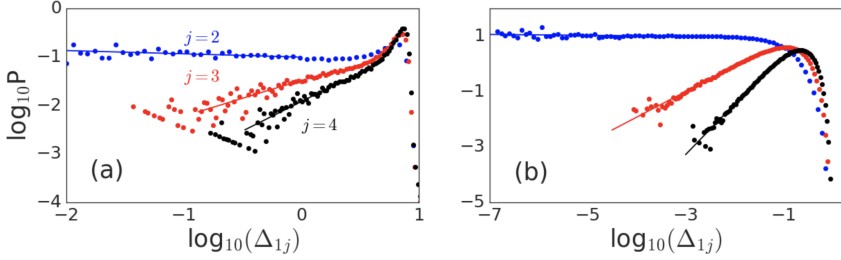

Figure 4: (a) Distribution of $\Delta_{1j}$ of NASNet-A trained on ImageNet. (b) Distribution of $\Delta_{1j}$ for logits that are sampled independently from a uniform random distribution for 5 million samples with 10 classes. $\Delta_{1j}$ of other models are in Appendix Fig. 14

Given the large density of small $\Delta_{12}$ values, we study an entropy penalty regularizer to make models more robust. Our proposed loss function can be written as: loss = old loss $- \lambda \sum_{i=1}^{n} p_i \log p_i$. where $\lambda$ is a hyperparameter, $n$ is the number of classes, and $p_i$ are the outputs of the neural network. This regularization term has been used by Miyato et al. (2017) for semi-supervised learning tasks. It aims to increase the confidence of the network on each sample, which is the opposite of previous regularization attempts that penalized confidence to increase generalization accuracy (Pereyra et al.,

2017; Szegedy et al., 2015). By penalizing the entropy of the softmax outputs, we aim to increase the logit differences.

In Fig. 5, we show that entropy regularization with a $\lambda = 4.5$ increases the adversarial robustness both for regularly trained networks and step l.l. adversarially trained networks, and both for permutation invariant and regular MNIST. We note that the same qualitative results hold for other values of $\lambda$ we tried. Despite the increase in adversarial accuracy, the permutation invariant MNIST model has 0.8% lower clean accuracy when trained with the entropy penalty. In Appendix Fig. 11, we show that a wide ResNet trained with the entropy regularizer has improved robustness with no loss in clean accuracy.

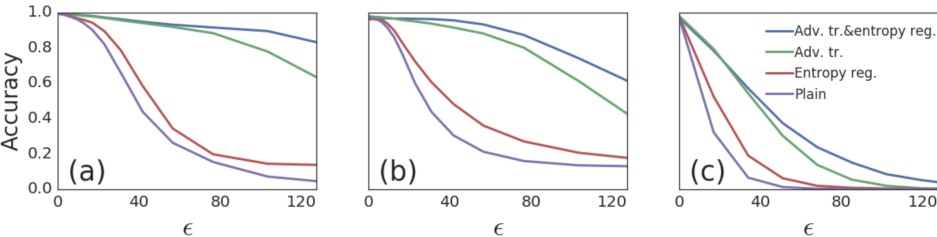

Figure 5: Step l.l. attack adversarial accuracy as a function of $\epsilon$ for CNN and permutation invariant MNIST in (a) and (b), respectively. Regular training (purple), entropy regularization (red), adversarial training (green), and adversarial training with entropy regularization (blue) have been implemented. Adversarial training was done using the step l.l. method. In (c), we show the PGD attack adversarial accuracy on permutation invariant MNIST trained with and without step l.l. adversarial training.

We investigate whether the increased adversarial robustness is due to increased logit differences. In Fig. 6, we plot the distribution of $\Delta_{1j}$ for $j$ up to 4, for two networks trained on permutation invariant MNIST, with and without entropy regularization. As expected, margins are shifted to larger values and density of samples with small $\Delta_{1j}$ are reduced. The tails still follow a power-law form with the same exponents, however there are fewer samples with small margins compared to a regularly trained network. Although entropy regularization made our networks white-box attacks, it did not lead to a significant improvement against black-box attacks. For this reason, we focus on the influence of network architectures on adversarial sensitivity below.

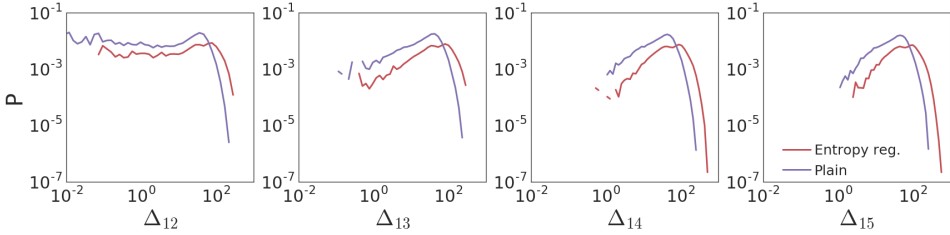

Figure 6: Distribution of $\Delta_{1j}$ up to $j = 5$ for permutation invariant MNIST trained with and without entropy regularization in red and purple, respectively.

## 4 ROLE OF NETWORK ARCHITECTURES IN ADVERSARIAL ROBUSTNESS

Several recent papers observe that larger networks are more robust against adversarial examples, regardless if they are adversarially trained or not (Kurakin et al., 2016b; Madry et al., 2017). However, it is not clear if network architectures play an important role in adversarial robustness. Are larger models more robust because they have more trainable parameters, or simply because they have higher clean accuracy? Is it possible to find more robust network architectures that do not necessarily have more parameters? We run several experiments to answer these questions, as well

as to find an adversarially more robust model on CIFAR10. We perform neural architecture search (NAS) with reinforcement learning. Our search space and procedure are almost exactly the same as in Zoph et al. (2017). One difference is that we restrict the search space so that the normal cell must be the same as the reduction cell. This reduces the complexity of the search space as now we have only half as many predictions. Finally, we increase the number of prediction steps from 5 to 7 to slightly gain back the complexity that was lost when we restricted the normal cell to be equal to the reduction cell.

We carry out two experiments:

- Experiment 1: NAS where child models are trained with clean and step l.l. adversarial examples and the reward is computed on the validation set with FGSM adversarial accuracy at $\epsilon = 8$.

- Experiment 2: NAS where child models are trained with clean and PGD adversarial examples and the reward is computed on the validation set with FGSM adversarial accuracy at $\epsilon = 8$.

In both experiments, child models are trained for 10 epochs on a training set of 25 thousand samples. Child models are trained on mini-batches where half of the samples are adversarially perturbed, following the procedure in (Kurakin et al., 2016b). At the end of each experiment, we pick the child model with the highest FGSM adversarial accuracy at $\epsilon = 8$ on the validation set of 5000 samples, and scale up the number of filters. We train the enlarged models for 100 epochs on the full training set of 45000 samples for 12 different hyperparameter sets, and pick the one with the highest adversarial accuracy on the validation set. Finally, we report below the performance of these models on a held-out test set of 10 thousand samples. To provide a comparison with the results of our two experiments, we also run a vanilla NAS where the reward is clean validation accuracy. We will refer to the best architecture from vanilla NAS as NAS Baseline. When trained using the setup above only on clean examples, NAS Baseline reaches a test set accuracy of 95.3%.

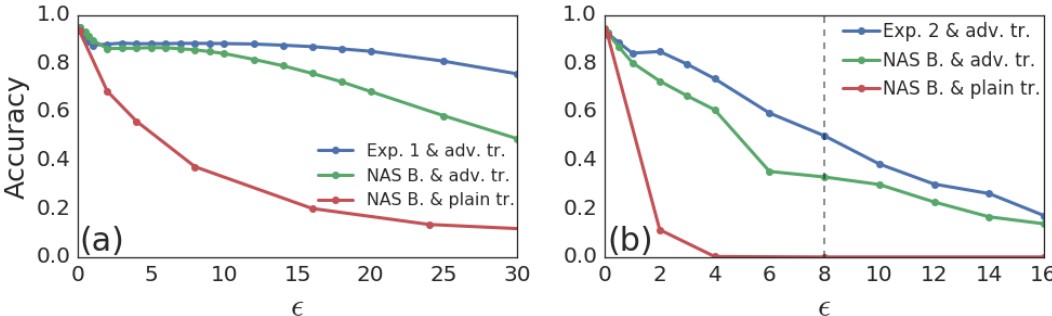

Figure 7: (a) step l.l. adversarial accuracy of NAS Baseline trained with and without step l.l. adversarial examples in green and red, respectively. Best model from Experiment 1 is shown in blue. (b) PGD adversarial accuracy of NAS Baseline trained with and without PGD adversarial examples in green and red, respectively. Best model from Experiment 2 is shown in blue.

We present the results of Experiment 1 in Fig 7(a). Here the green curve is the adversarial accuracy NAS Baseline. The blue curve is the adversarial accuracy of a network architecture that was found by Experiment 1. Both of these architectures are trained with the same adversarial training procedure. We try the same sets of hyperparameters and report here the models with best adversarial accuracy at $\epsilon = 8$ on the validation set. Adversarial training reduced the clean accuracy by 0.2%. Adversarially trained models both have clean accuracy of 95.1% on the test set, whereas the model that was trained without adversarial training reached 95.3% accuracy.

We next use PGD adversarial examples in the training of child models, to find architectures that are more robust to any adversarial attack within an $\epsilon$ ball (Madry et al., 2017). Following the training procedure by Madry et al. (2017), we use 7 steps of size 2, for a total $\epsilon = 8$. We present the results of Experiment 2 in Fig. 7(b). As was the case in Experiment 1, the architecture found by adversarial NAS leads to a more robust model. At $\epsilon = 8$, the architecture from Experiment 2

| | Clean | White-box | | | Black-box | | |
|---|---|---|---|---|---|---|---|
| | | FGSM | step l.l. | PGD | FGSM | step l.l. | PGD |
| Madry et al. (2017) | 87.3% | 56.1% | - | 50.0% | 67.0 % | - | 64.2% |
| This work | **93.2**% | **63.6**% | 77.9% | 50.1% | **78.1** % | 84.9% | **75.0**% |

Table 1: Performance of our best architecture from Experiment 2 at $\epsilon = 8$. Black-box attacks are sourced from a copy of the network independently initialized and trained.

reaches a 17% higher adversarial accuracy on PGD examples. We compare our results to the results by Madry et al. (2017). Madry et al. (2017) trained only on PGD examples, whereas half of our minibatches are clean examples. Despite this, we match their accuracy on white-box PGD attacks. Against other white- and black-box attacks our model is more robust, and our clean accuracy is 5.9% higher. We also note that NAS Baseline model has 4.9 million trainable parameters, whereas the model from Experiments 1 and 2 have 2.3 million and 3.5 million parameters, respectively. NAS found an adversarially more robust architecture with many fewer parameters. Best architecture from Experiment 2 and NAS Baseline are presented in Appendix Fig. 16. Finally, we study the

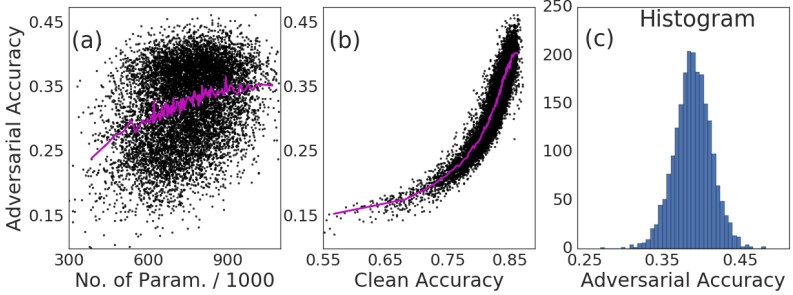

Figure 8: Performance of child models on the validation set. FGSM adversarial accuracy at $\epsilon = 8$ vs. number of trainable parameters and clean accuracy in (a) and (b), respectively. Black dots represent each child model, purple line is a running average. Figure (c) is the histogram of the adversarial accuracy for models with clean accuracy larger than 85%.

performance statistics of child models during NAS. In Fig. 8, we report the results for 9360 child models that were trained during Experiment 1. As explained above, these models are only trained for 10 epochs. In Fig. 8(a), we see that the correlation between adversarialy accuracy and the number of trainable parameters of the model is not very strong. On the other hand, adversarial accuracy is strongly correlated with clean accuracy (Fig. 8(b)). We hypothesize that this is the reason both Madry et al. (2017) and Kurakin et al. (2016b) found that making networks larger increased adversarial robustness, because it also increased the clean accuracy. This implies that commonly used architectures, like Inception v3 and ResNet, benefit from having more parameters. This however was not the case for most child models during NAS. On the other hand, having a high clean accuracy is not sufficient for adversarial robustness. As seen in Fig. 8(c), there is a large variance in the adversarial accuracy of models with good clean accuracy. The range of adversarial accuracies in the histogram of models with larger than 85% clean accuracy is 22% and the standard deviation is 2.6%. For this reason, our experiments led to more robust architectures than NAS Baseline.

## 5 CONCLUSION

In this paper we studied common properties of adversarial examples across different models and datasets. We theoretically derived a universality in logit differences and adversarial error of machine learning models. We showed that architecture plays an important role in adversarial robustness, which correlates strongly with clean accuracy.

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

# 6 APPENDIX

## 6.1 FURTHER EXPERIMENTS

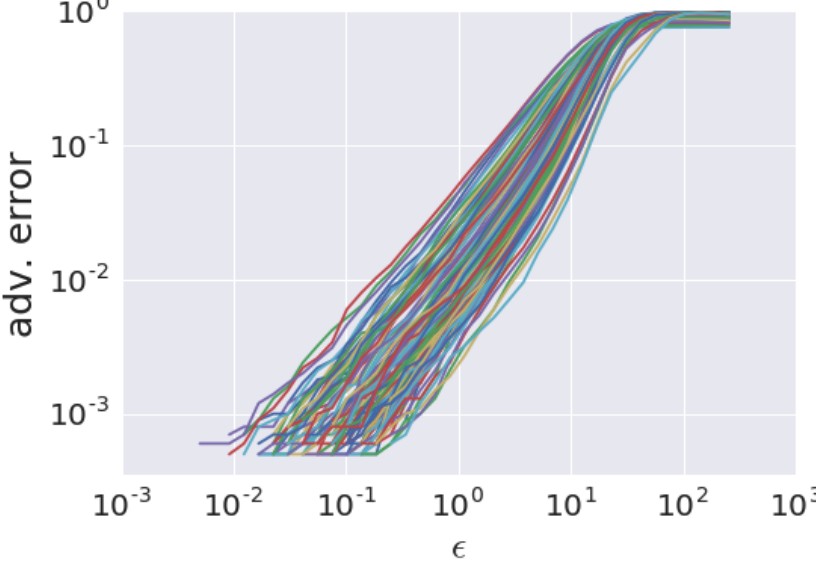

Figure 9: Adversarial error for hundreds of models trained on MNIST, including fully-connected and convolutional models. We only show models with clean accuracy larger than 80%.

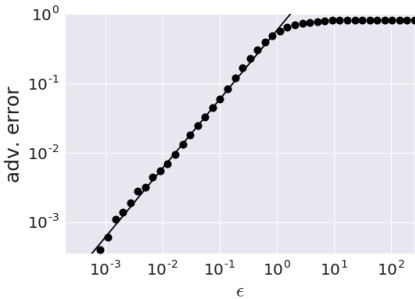

Figure 10: Points represent the adversarial error due to FGSM as a function of $\epsilon$ for a 32-layer ResNet trained on CIFAR10. Straight line is a power law fit with an exponent of 0.99.

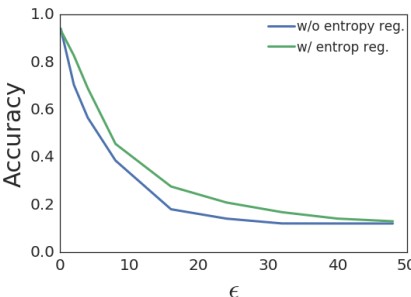

Figure 11: Effect of adding an entropy regularization ($\lambda = 3.0$): step l.l. adversarial accuracy of wide ResNet on CIFAR10, with and without entropy regularization. Both models have a clean accuracy of $94\%$. They were both trained for 100 epochs with the same hyperparameters.

## 6.2 DERIVATIONS

Here we derive several of the results found in the main text.

### 6.2.1 LINEAR RESPONSE

First, we compute the linear response of the network to the $L_2$ FGSM attack. Let, $f : \mathbb{R}^N \to \mathbb{R}^M$ be a neural network (or other model) structured such that $f = \text{softmax} \circ h$ where $h : \mathbb{R}^N \to \mathbb{R}^M$ maps inputs to logits. Additionally define a loss $L : \mathbb{R}^M \to \mathbb{R}$ which can be cross-entropy, $L^2$, etc.. To generate adversarial examples we start with an input $x \in \mathbb{R}^N$ and a corresponding target $t \in \mathbb{R}^M$ such that $t_\beta = 1$ if $\beta = \gamma$ for some $\gamma$ and $t_\beta = 0$ otherwise. We assume our network gets the answer correct so that $h_\gamma > h_\beta$ for all $\beta \neq \gamma$. Then we apply the adversarial perturbation,

$$x'_\alpha = x_\alpha + \epsilon \frac{\nabla_x L}{||\nabla_x L||_2}. \tag{11}$$

Note that we can write

$$\nabla_x L = \frac{\partial L}{\partial x_\alpha} = \sum_\beta \frac{\partial h_\beta}{\partial x_\alpha} \frac{\partial L}{\partial h_\beta} = \sum_\beta J_{\alpha\beta} \frac{\partial L}{\partial h_\beta} = J\delta. \tag{12}$$

Where we associate $J_{\alpha\beta} = \partial h_\beta / \partial x_\alpha$ with the input-to-logit Jacobian linking the inputs to the logits and $\delta = \partial L / \partial h_\beta$ the error of the outputs of the network.

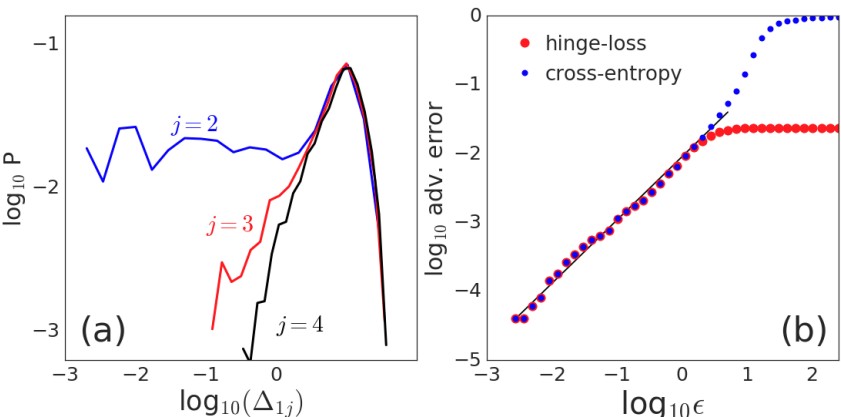

Figure 12: Fully connected network trained on MNIST with hinge loss. (a) Distributions of logit differences. (b) Red dots represent the adversarial error when FGSM attack uses the same hinge loss from training. Blue dots represent the adversarial error when FGSM attack uses a cross-entropy loss to create the adversarial examples.The line is a power-law fit with an exponent of 0.98

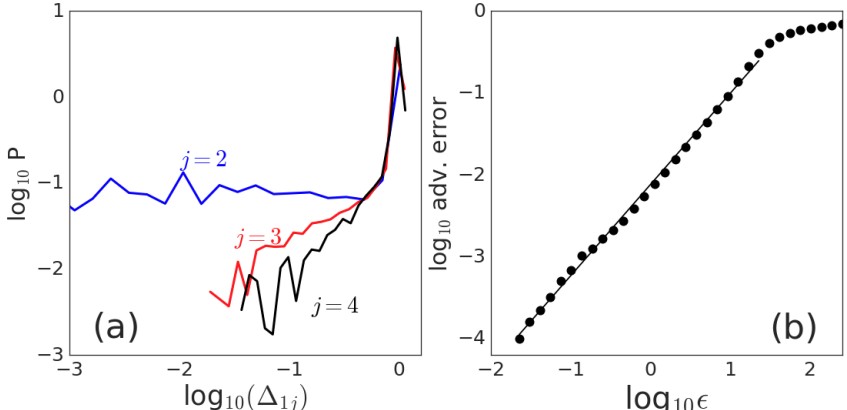

Figure 13: Fully connected network trained on MNIST with L2-norm loss. (a) Distributions of logit differences. (b) Black dots represent the adversarial error due to FGSM. The line is a power-law fit with an exponent of 1.02

We can compute the change to the logits of the network due to this perturbation. We find,

$$h(x') = h(x + \epsilon \nabla_x L / ||\nabla_x L||_2) \tag{13}$$

$$h'_\beta \approx h_\beta + \frac{\epsilon}{||\nabla_x L||_2} \sum_\alpha \frac{\partial h_\beta}{\partial x_\alpha} \frac{\partial L}{\partial x_\alpha} + \mathcal{O}(\epsilon^2) \tag{14}$$

$$= h_\beta + \frac{\epsilon}{||\nabla_x L||_2} \sum_{\alpha\delta} \frac{\partial h_\beta}{\partial x_\alpha} \frac{\partial h_\delta}{\partial x_\alpha} \frac{\partial L}{\partial h_\delta} \tag{15}$$

where we have plugged in for eq. (11). Expressing the above equation in terms of the Jacobian, it follows that we can write the effect of the adversarial perturbation on the logits by,

$$h' = h + \epsilon \frac{J^T J \delta}{||J\delta||_2} \tag{16}$$

as postulated.

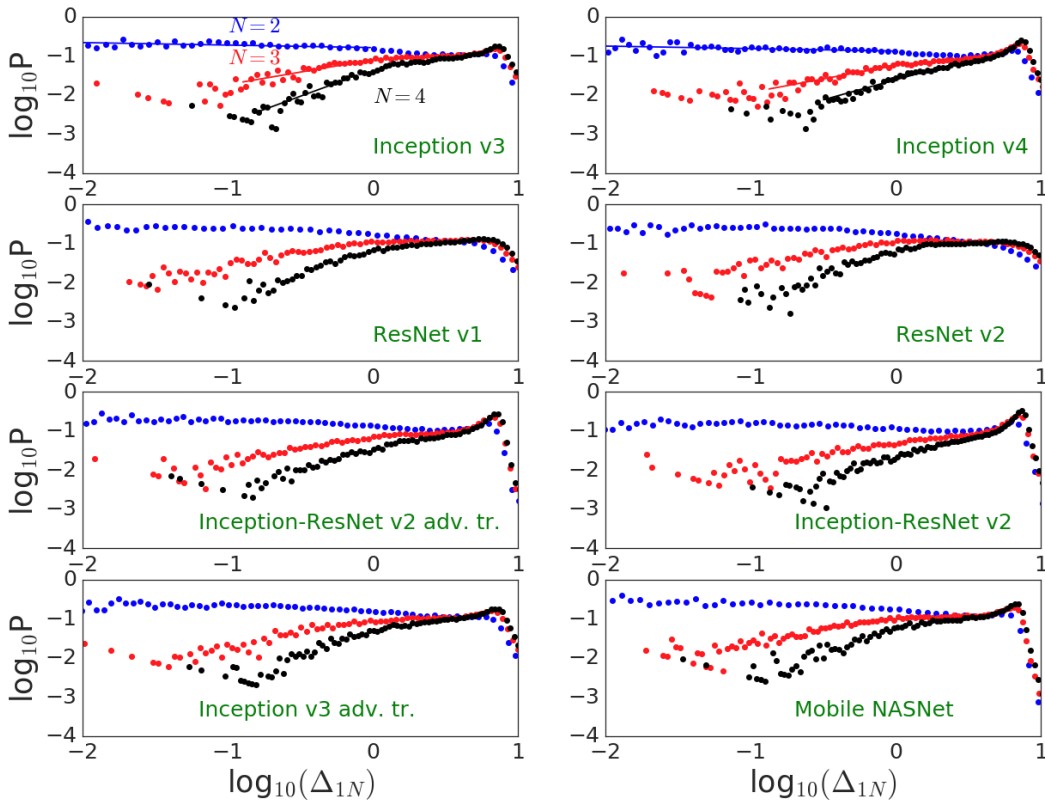

Figure 14: (a) Distribution of $\Delta_{1N}$ for other ImageNet models.

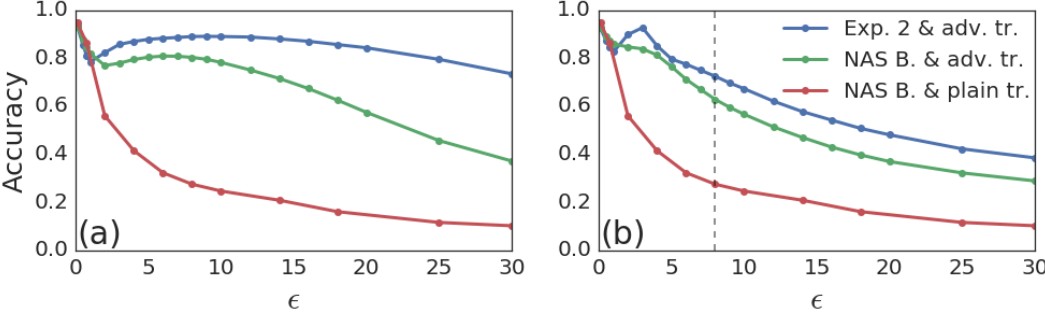

Figure 15: (a) FGSM adversarial accuracy of NAS Baseline trained with and without step l.l. adversarial examples in green and red, respectively. Best model from Experiment 1 is shown in blue. (b) FGSM adversarial accuracy of NAS Baseline trained with and without PGD adversarial examples in green and red, respectively. Best model from Experiment 2 is shown in blue.

### 6.2.2 UNIVERSAL PROPERTIES OF THE LOGIT DIFFERENCE DISTRIBUTION

We now show that

$$P(\Delta_{1j}) = C\Delta_{1j}^{j-2} + \mathcal{O}(\Delta_{1j}^{j-1}). \tag{17}$$

To make progress we will again make a mean field approximation and assume that each of the logits are i.i.d. with arbitrary distribution $P(h)$. We denote the cumulative distribution $F(h)$. While it is not obvious that the factorial approximation is valid here, we will see that the resulting distribution of $P(\Delta_{1j})$ shares many qualitative similarities with the distribution observed in real networks.

We first change variables from the logits to a sorted version of the logits, $r_i$. The ranked logits are defined such that $r_1 = \max(\{h_i\})$, $r_2 = \max(\{h_i\}\backslash\{r_1\})$, $\cdots$. Our first result is to compute the resulting joint distribution between $r_1$ and $r_j$,

$$P_j(r_1, r_j) = A(N, j)F^{N-j}(r_j)\left[F(r_1) - F(r_j)\right]^{j-2} P(r_j)P(r_1) \tag{18}$$

where $A(N, j) = N(N-1)\binom{N-2}{j-2}$ is a combinatorial factor. Eq. (18) has a simple interpretation. $F^{N-j}(r_j)$ is the probability that there are $N - j$ variables less than $r_j$; $[F(r_1) - F(r_j)]^{j-2}$ is the probability that $j - 2$ variables are between $r_j$ and $r_1$; $P(r_j)P(r_1)$ is the probability that there is one variable equal to each of $r_1$ and $r_j$. The combinatorial factor can be understood since there are $N$ ways of selecting $r_1$, $N - 1$ ways of selecting $r_j$, and $\binom{N-2}{j-2}$ ways of choosing $j - 2$ variables out of the remaining $N - 2$ to be between $r_j$ and $r_1$.

In terms of eq. (18) we can compute the distribution over $\Delta_{1j}$ to be given by,

$$P(\Delta_{1j}) = \int dr P_j(r + \Delta_{1j}, r) \tag{19}$$

$$= A(N, j) \int dr F^{N-j}(r) \left[F(r + \Delta_{1j}) - F(r)\right]^{j-2} P(r)P(r + \Delta_{1j}). \tag{20}$$

We can analyze this equation for small $\Delta_{1j}$. Expanding to lowest order in $\Delta_{1j}$,

$$P(\Delta_{1j}) \approx A(N, j) \int dr F^{N-j}(r) \left[F(r) + \Delta_{1j}P(r) - F(r)\right]^{j-2} P(r) \left[P(r) + \Delta_{1j}\frac{dP(r)}{dr}\right] \tag{21}$$

$$= A(N, j)\Delta_{1j}^{j-2} \int dr F^{N-j}(r)P^j(r) + \mathcal{O}(\Delta_{1j}^{j-1}). \tag{22}$$

Since the term in the integral does not depend on $\Delta_{1j}$ the result follows with,

$$C = N(N-1)\binom{N-2}{j-2} \int dr F^{N-j}(r)P^j(r). \tag{23}$$

### 6.2.3 ARCHITECTURES

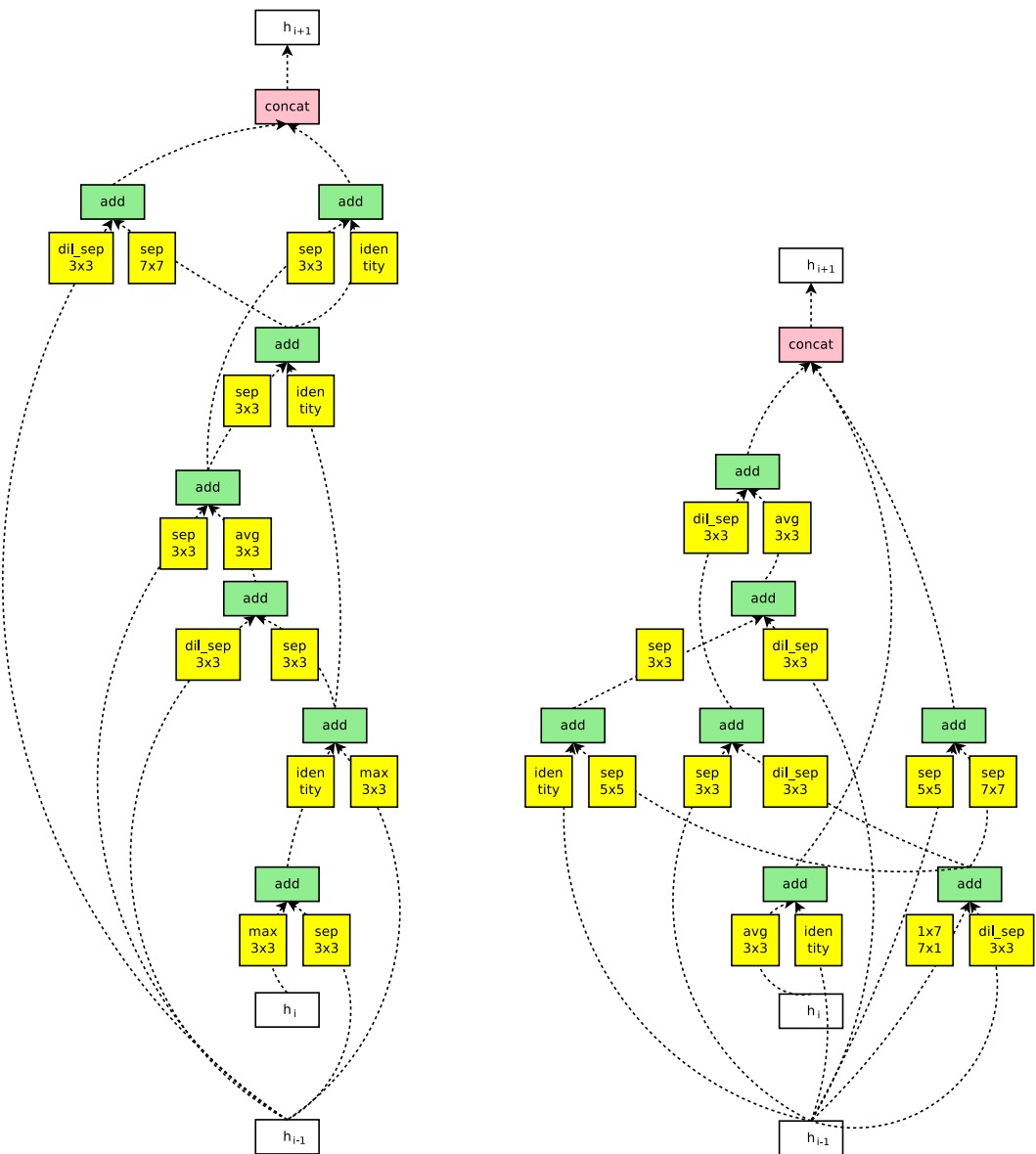

Figure 16: Left: Best architecture from Experiment 1. Right: Architecture of NAS Baseline. We note that the architecture from Experiment 1 is "longer" and "narrower" than previous architectures found by NAS for higher clean accuracy (Zoph & Le, 2016; Zoph et al., 2017).

