# OpenReview forum: "Intriguing Properties of Adversarial Examples"
_ICLR.cc/2018/Conference — Invite to Workshop Track_

### Official Review · AnonReviewer1 · 2017-11-25
**Their explanation seems to be done by non-strict argument,  and their proposed methods do not seem related to their discovery so much.**

**Rating:** 5
**Confidence:** 2

**Review:**

This paper insists that adversarial error for small adversarial perturbation follows power low as a function of the perturbation size, and explains the cause by the logit-difference distributions using mean-field theory.
Then, the authors propose two methods for improving adversarial robustness (entropy regularization and NAS with reinforcement learning).

[strong points]
* Based on experimental results over a broad range of datasets, deep network models and their attacks.
* Discovery of the fact that adversarial error follows a power low as a function of the perturbation size epsilon for small epsilon.
* They found entropy regularization improves adversarial robustness.
* Their neural architecture search (NAS) with reinforcement learning found robust deep networks.

[weak points]
* Unclear derivation of Eq. (9). (What expansion is used in Eq. (21)?)
* Non-strict argument using mean-field theory.
* Unclear connection between their discovered universality and their proposals (entropy regularization and NAS with reinforcement learning).

---

> ### Author Response · Authors · 2017-12-13
> **We clarified the expansion in Eq. 21, and would be happy to address any other step if needed.**
>
> We thank the reviewer for a careful reading of the manuscript and the helpful feedback. We are glad that the reviewer found several strong points about our paper. Below we respond point by point to the criticism:
>
> 1) In Eq. (21) we expand both F(r+\Delta_{1j}) and P(r+\Delta_{1j}) to lowest order in \Delta_{1j}, using regular Taylor expansion. We have added another step to the derivation to clarify.
>
> 2) While mean field theory is an approximate framework, it has a long history of effective use across a wide range of fields studying complex behavior including machine learning. For example, there are papers that approach neural networks from a mean field perspective dating back to at least 1989 [1]. Here, at each step of our calculation we evaluate the validity of the mean field approximation in Fig. 3 and Fig. 4. If there is a specific point in the approximation that the reviewer objects to, we would be happy to address it further.
>
> 3) Our proposed entropy regularization is directly related to the finding that logit difference distribution is universal. Since the adversarial error has a universal form due to the universal behavior of logit difference distribution, we tried to increase the logit differences to make our models more robust. As we show in Fig. 6, the entropy regularizer does increase the logit differences, as expected. Due to the increased logit differences, models that were trained with and without adversarial training are more robust to adversarial examples, as shown in Fig. 5 (MNIST) and Fig. 11 (CIFAR10).
>
> As mentioned in the paper, although the functional form of the adversarial error is universal, better models are quantitatively more robust to adversarial examples (e.g. Figs 1a and 1b). Given this, we wanted to study whether architecture can be engineered to improve adversarial accuracy. As mentioned in our submission, recent papers have found that larger models are more robust, but left unanswered whether models that generalize better are less susceptible to adversarial examples [2,3]. Using NAS, we show that models that generalize better are more robust, however model size does not seem to correlate strongly with adversarial sensitivity. Our findings together present a unified analysis of a model’s sensitivity to adversarial examples: commonalities among datasets, cause of the commonalities, and dependence on architecture.
>
> [1] Peterson, Carsten. "A mean field theory learning algorithm for neural networks." Complex systems 1 (1987): 995-1019.
> [2] Kurakin, Alexey, Ian Goodfellow, and Samy Bengio. "Adversarial machine learning at scale." arXiv preprint arXiv:1611.01236 (2016).
> [3] Madry, Aleksander, et al. "Towards deep learning models resistant to adversarial attacks." arXiv preprint arXiv:1706.06083 (2017).

---

### Official Review · AnonReviewer2 · 2017-11-26

**Rating:** 8
**Confidence:** 3

**Review:**

Very intriguing paper and results to say the least. I like the way it is written, and the neat interpretations that the authors give of what is going on (instead of assuming that readers will see the same). There is a well presented story of experiments to follow which gives us insight into the problem.

Interesting insight into defensive distillation and the effects of uncertainty in neural networks.

Quality/Clarity: well written and was easy for me to read
Originality: Brings both new ideas and unexpected experimental results.
Significance: Creates more questions than it answers, which imo is a positive as this topic definitely deserves more research.

Remarks:
- Maybe re-render Figure 3 at a higher resolution?
- The caption of Figure 5 doesn't match the labels in the figure's legend, and also has a weird wording, making it unclear what (a) and (b) refer to.
- In section 4 you say you test your models with FGSM accuracy, but in Figure 7 you report stepll and PGD accuracy, could you also plot the same curves for FGSM?
- In Figure 4, I'm not sure I understand the right-tail of the distributions. Does it mean that when Delta_ij is very large, epsilon can be very small and still cause an adversarial pertubation? If so does it mean that overconfidence in the extreme is also bad?

---

> ### Author Response · Authors · 2017-12-14
> **Typos corrected and figures added**
>
> We thank the reviewer for a careful reading of the manuscript and the helpful suggestions.
> We are delighted that the reviewer thinks this topic deserves more research; we certainly agree!
>
> We implemented the suggestions by the reviewer, as detailed below:
> - Re-rendered Fig. 3 at a higher resolution. We noticed that Fig. 3 may look pixelated on certain web browsers, but rendered correctly on all pdf viewers we have tried.
> - Corrected the typos and clarified the caption of Fig. 5. We appreciate the reviewer noticing this.
> - Experiment 1 networks were trained with stepll and Experiment 2 networks were trained with PGD. However, we did use FGSM accuracy on the validation set to choose the architectures. For this reason, we followed the reviewer's suggestion and plotted the same curves for FGSM attack in Figure 15.
> - Fig. 4 presents histograms where both axes are shown in log scale. The right-tail of the distributions signify that there are not many samples with as large \Delta_{ij} values.

---

### Official Review · AnonReviewer3 · 2017-11-29
**Interesting experiments, wrong conclusions.**

**Rating:** 3
**Confidence:** 4

**Review:**

This work presents an empirical study aiming at improving the understanding of the vulnerability of neural networks to adversarial examples. Paraphrasing the authors, the main observation of the study is that the vulnerability is due to an inherent uncertainty that neural networks have about their predictions ( the difference between the logits). This is consistent across architectures, datasets. Further, the authors note that "the universality is not a result of the specific content of these datasets nor the ability of the model to generalize."

While this empirical study contains valuable information, its above conclusions are factually wrong. It can be theoretically proven at least using two routes. They are also in contradiction with other empirical observations consistent across several previous studies.

1-Constructive counter-argument: Consider a neural network that always outputs a constant prediction. It (1) is by definition independent of any dataset (2) generalizes perfectly (3) has zero adversarial error, hence contradicting the central statement of the paper.

2- Analysis-based counter-argument: Consider a neural network with one hidden layer and two classes. It is easy to show that the difference between the scores (logits) of the two classes is linear in the operator norm of the hidden weight matrix and linear in the L2-norm of the last weight vector. Therefore, the robustness of the model indeed depends on its capability to generalize because the latter is essentially governed by the geometric margin of the linear separator and the spectral norm of the weight matrix (see [1,2,3]). QED.

3- Further, the lack of calibration of neural networks and its causes are well known. Among other things, it is due to the use of building blocks (such as batch-norm [4]), regularization (e.g., weight decay) or the use of softmax+cross-entropy during training. While this is convenient for optimization reasons, it indeed hurts the calibration. The authors should try to train a neural network with a large margin criteria and see if the same phenomenon still holds when they measure the geometric margin. Another alternative is to use a temperature with the softmax[4]. Therefore, the observations of the empirical study cannot be generalized to neural networks and should be explicitly restricted to neural networks using softmax with cross-entropy as criteria.

I believe the conclusions of this study are misleading, hence I recommend to reject the paper.


[1] Spectrally Normalized Margin-bounds Margin bounds for neural networks (Bartlett et al., 2017)
[2] Parseval Networks: Improving Robustness to Adversarial Examples (Cisse et al., 2017)
[3] Formal Guarantees on the Robustness of a classifier against adversarial examples (Hein et al., 2017)
[4] On the Calibration of Modern Neural Networks (Guo et al., 2017)

---

> ### Author Response · Authors · 2017-12-13
> **Experiments suggested by reviewer corroborate our original conclusions, we reworded ambiguous sentences in the abstract**
>
> We would first like to thank the reviewer for their careful reading of our manuscript and thoughtful comments. We are glad that the reviewer believes our study contains valuable information and interesting experiments. Meanwhile, we would like to address the concerns raised.
>
> Summary: We are confident that our results and conclusions are not at odds with the perspective of the referee. We believe the main issue stems from some ambiguous language in the original text that we have now corrected. We have also implemented the additional experiments proposed by the referee and have found them to corroborate our original conclusions.
>
> Details: We believe that there is some confusion regarding what was meant by our statement “the universality is not a result of the specific content of these datasets nor the ability of the model to generalize." We are not proposing that the susceptibility of a neural network to adversarial examples is independent of its ability to generalize. Instead, we are saying that the functional form of the adversarial error as a function of epsilon does not depend on generalization (i.e. that it should scale like A * \epsilon regardless of the network’s ability to generalize, as shown by our experiments on randomly sampled logits and MNIST with randomly-shuffled labels). In fact, we agree with the referee that the constant, A, will depend on the spectral norm of the Jacobian (and hence the readout weight matrix) and on the network’s ability to generalize. We copy some excerpts from the original submission to corroborate this below. However, the reviewer’s concerns allowed us to realize that our original phrasing was ambiguous. We have therefore reworded our conclusions to be clearer by replacing the problematic statement with: “Here we show that distributions of logit differences have a universal functional form. This functional form is independent of architecture, dataset, and training protocol; nor does it change during training.” We have also removed the sentence “Here we argue that the origin of adversarial examples is primarily due to an inherent uncertainty that neural networks have about their predictions.”
>
> Excerpts from the original text showing agreement with the referee:
>
> “We observe that although the qualitative form of logit differences and adversarial error is universal, it can be quantitatively improved with entropy regularization and better network architectures.”
> “...vanilla NASNet-A (best clean accuracy in our study) has a lower adversarial error than adversarially trained [models]…”
> In eq. 8 we find that the threshold for an adversarial error is proportional to J^TJ. This is clearly proportional to the spectral norm of the Jacobian.

---

> > ### Author Response · Authors · 2017-12-13
> > **continuation**
> >
> > Responses to specific points:
> > 1- This thought experiment actually agrees with our paper: a neural network that always outputs a constant has no uncertainty about its predictions, and thus has zero adversarial error. Furthermore, our theory assumed uncorrelated logits, but we empirically show that the power-law tails are robust to the the amount of correlation present in the logits of the commonly used neural networks. In the thought experiment suggested by the reviewer, the logits are maximally correlated. It is for this reason that our theory may not apply. Finally, we note that in this example the input-logit Jacobian is zero. In this case, our mathematical framework correctly predicts that \hat\epsilon\to\infty and so no amount of adversarial perturbation will change the predicted class.
> >
> > 2- As mentioned above, we agree that models that generalize better have higher adversarial robustness. As can be seen in Fig. 1a and 1b, the models with best generalization (NASNet, Inception-ResNet v2, Inception v4) are also adversarially most robust, especially for small epsilon values. This is why we performed Neural Architecture Search to find adversarially robust architectures: although the qualitative form of the adversarial error is a power-law with similar exponents, the quantitative robustness can be improved via adversarial training, architecture engineering, and regularization. We have used all three of these techniques to increase the adversarial robustness in our study.
> >
> > 3- The lack of calibration of neural networks and its causes may be well known, but our contribution is to point out that the functional form of the logit differences is universal across datasets and models, and unchanged after training.
> >
> > We added two new figures to the appendix, Fig. 12 and Fig. 13, which show that the reported universality is not restricted to neural networks using softmax with cross-entropy as loss. In Fig. 12, we trained a fully-connected network on MNIST with hinge-loss (as suggested by the reviewer). We attacked this network both by differentiating the hinge-loss and a cross-entropy loss (attacks with cross-entropy loss are more successful, as also observed in the submission “Certified Defenses against Adversarial Examples”). We show that both of these attacks lead to the same universal behavior, both for adversarial error and for logit differences. We repeat the same experiment using an L2-norm loss for training, and reach the same results as the experiments in the original submission.
> >
> > In short, our original submission is in agreement with the reviewer’s perspective; and newly performed experiments as suggested by the reviewer obey the universality that is presented by our paper.

---

### Public Comment · (anonymous) · 2017-11-09
**Robustness to very small perturbations**

Very cool insights, I really enjoyed your paper. I had a question about your experiments with FGSM attacks for small epsilon (Figure 1 & 2). What is the rationale for considering non-integer values of epsilon here (especially epsilon < 1), since the resulting perturbed inputs do not actually represent valid RGB images? As I understand it, simply converting the image to a valid RGB representation would remove any perturbation with epsilon < 0.5.
While it is interesting that adversarially trained models did not learn to be robust in that regime, is that really surprising given that Kurakin et al. (2016) seem to only consider integer values of epsilon in their paper?

---

> ### Author Response · Authors · 2017-11-14
> **Adversarial training did include fractional epsilon, and unit L2-norm attacks lead to larger pixel changes.**
>
> Thanks for the positive comment and the interesting question. Kurakin et al. did use non-integer values of epsilon during training. As mentioned in their paper, epsilon was sampled from a truncated normal defined in [0,16].
>
> Regarding test-time: as we show in Fig 2c for MNIST, attacks with unit L2 norm have the same power-law form and exponent as FGSM, but allow for much larger change in each pixel value. For ImageNet, unit L2 norm attack has the same power-law form and exponent up to an epsilon of 70; this means that one pixel could change by as large as 70 due to adversarial distortion and still be in the power-law region. We will include an additional plot about this in the next version of our submission.

---

### Author Response · Authors · 2018-01-04
**summary of our previous rebuttals**

We thank the reviewers for their reviews. We would like to summarize our responses to individual reviewers. Our work shows two fundamental (and surprising) commonalities across datasets and models: logit differences and adversarial error have the same functional form across all models tested on MNIST, CIFAR10, and ImageNet. We show that these commonalities even hold for random data, and we theoretically derive the origin and its consequences under a mean-field approximation. Based on our observations we propose a counter-intuitive regularization term, entropy penalty, to reduce adversarial sensitivity. Since our results imply that better models are more robust, we use neural architecture search (NAS) to find a model that is adversarially more robust than previously available models. We can move the part on NAS to appendix if the reviewers see fit. In summary, our paper makes important contributions on three fronts: empirical findings, theoretical explanations of these findings, and practical results on adversarial robustness.

Main criticism by AnonReviewer3 is based on a miscommunication. As explained below, our results agree with this reviewer: models that generalize better tend to be more robust. Furthermore, we implemented the experiments proposed by this reviewer, including a thought experiment. We show that the results of all of these experiments support our conclusions. Thanks to the suggestions by this reviewer, we have changed the wording to make our point more clear.

AnonReviewer1 is concerned with the mean field approximation we employed, which we disagree with. Mean-field approximation has been used for more than a century to model complex systems, and its strengths as well as shortcomings are well understood[1,2,3,4,5]. We evaluated the validity of our approximations at every step. We are happy to discuss any particular step of the derivation, however we believe that the reviewer’s general criticism of mean-field approximation is not specific enough to warrant the rejection of the paper or for us to address the concern. The step of the derivation that the reviewer found unclear (Eq. 21) was just a Taylor expansion to smallest order, which we clarified in our revision.

Overall, the reviewer reports do not have concrete disagreements with our results, and the reviewers found our experiments to be interesting over a broad range of datasets, models, and attacks. We have supported our arguments with concrete empirical evidence. In light of these, we hope that the AnonReviewer1 and AnonReviewer3 reconsider their scores.

[1] Weiss, Pierre. "L'hypothèse du champ moléculaire et la propriété ferromagnétique." J. phys. theor. appl. 6.1 (1907): 661-690.
[2] Peterson, Carsten. "A mean field theory learning algorithm for neural networks." Complex systems 1 (1987): 995-1019.
[3] Kardar, Mehran. Statistical physics of fields. Cambridge University Press, 2007.
[4] Poole, Ben, et al. "Exponential expressivity in deep neural networks through transient chaos." Advances in neural information processing systems. 2016.
[5] Schoenholz, Samuel S., et al. "Deep Information Propagation." ICLR. 2017.

---

### Decision · Program_Chairs · 2018-01-29
**ICLR 2018 Conference Acceptance Decision**

**Decision:**

Invite to Workshop Track

**Comment:**

I am somewhat of two minds from the paper. The authors show empirically that adversarial perturbation error follows power law and looks for a possible explanation. The tie in with generalization is not clear to me and makes me wonder how to evaluate the significance of the finding of the power law distribution..  On the other hand, the authors present an interesting analysis, show that the finding holds in all the cases they explored and also found that architecture search can be used to find neural networks that are more resilient to adversarial search (the last shouldn't be surprising if that was indeed the training criterion).

All in all, I think that while the paper needs a further iteration prior to publication, it already contains interesting bits that could spur very interesting discussion at the Workshop.

(Side note: There's a reference missing on page 4, first paragraph)